# Microgravity Modifies the Phenotype of Fibroblast and Promotes Remodeling of the Fibroblast–Keratinocyte Interaction in a 3D Co-Culture Model

**DOI:** 10.3390/ijms23042163

**Published:** 2022-02-16

**Authors:** Valeria Fedeli, Alessandra Cucina, Simona Dinicola, Gianmarco Fabrizi, Angela Catizone, Luisa Gesualdi, Simona Ceccarelli, Abdel Halim Harrath, Saleh H. Alwasel, Giulia Ricci, Paola Pedata, Mariano Bizzarri, Noemi Monti

**Affiliations:** 1Department of Experimental Medicine, Sapienza University of Rome, Viale Regina Elena 324, 00161 Rome, Italy; valeria.fedeli@uniroma1.it (V.F.); simona.ceccarelli@uniroma1.it (S.C.); mariano.bizzarri@uniroma1.it (M.B.); 2Systems Biology Group Laboratory, Sapienza University, 00161 Rome, Italy; simonadinicola.sd@gmail.com (S.D.); gianmarco1289@yahoo.it (G.F.); 3Department of Surgery “Pietro Valdoni”, Sapienza University of Rome, Via Antonio Scarpa 14, 00161 Rome, Italy; alessandra.cucina@uniroma1.it; 4Policlinico Umberto I, Viale del Policlinico 155, 00161 Rome, Italy; 5Section of Histology and Embryology, Department of Anatomy, Histology, Forensic Medicine and Orthopedics, Sapienza University of Rome, Viale Regina Elena 336, 00161 Rome, Italy; angela.catizone@uniroma1.it (A.C.); luisa.gesualdi@uniroma1.it (L.G.); 6Department of Zoology, College of Science, King Saud University, Riyadh 11451, Saudi Arabia; halim.harrath@gmail.com (A.H.H.); salwasel@ksu.edu.sa (S.H.A.); 7Department of Experimental Medicine, University of Campania Luigi Vanvitelli, 80138 Naples, Italy; giulia.ricci@unicampania.it; 8Department of Medicine, University of Salerno, Via Giovanni Paolo II 132, 84084 Salerno, Italy; papedata@unisa.it

**Keywords:** microgravity, fibroblasts, myofibroblasts, cytoskeleton, α-SMA, keratinocytes, co-culture

## Abstract

Microgravity impairs tissue organization and critical pathways involved in the cell–microenvironment interplay, where fibroblasts have a critical role. We exposed dermal fibroblasts to simulated microgravity by means of a Random Positioning Machine (RPM), a device that reproduces conditions of weightlessness. Molecular and structural changes were analyzed and compared to control samples growing in a normal gravity field. Simulated microgravity impairs fibroblast conversion into myofibroblast and inhibits their migratory properties. Consequently, the normal interplay between fibroblasts and keratinocytes were remarkably altered in 3D co-culture experiments, giving rise to several ultra-structural abnormalities. Such phenotypic changes are associated with down-regulation of α-SMA that translocate in the nucleoplasm, altogether with the concomitant modification of the actin-vinculin apparatus. Noticeably, the stress associated with weightlessness induced oxidative damage, which seemed to concur with such modifications. These findings disclose new opportunities to establish antioxidant strategies that counteract the microgravity-induced disruptive effects on fibroblasts and tissue organization.

## 1. Introduction

Research in space biomedicine is regaining momentum as space agencies, both in the USA and Europe, plan an ambitious program called *ARTEMIS*. Its primary goal is to send humans back to the Moon by 2025 [1]. However, a number of biomedical threats actually challenge the sustainability of prolonged exposure to the outer space environment [2]. Cosmic rays, radiation, disruption of biological rhythms, and microgravity can cause adverse effects upon several physiological functions, thus impairing the immune system, bone density, as well as neurological and cardiovascular function [3,4,5]. Specifically, weightlessness can impair human physiology by acting at the systemic level and by affecting a bewildering number of critical and genomic pathways within cells and tissues [6,7]. Several lines of evidence suggest that immune, muscular, bone, breast, and many other cells significantly change their morphology and dynamical behavior during microgravity exposure [8]. It is worth noting that epithelial cells—in both normal and pathological conditions—dynamically interact with the extracellular matrix (ECM) and a wide array of mesenchymal cells, including fibroblasts. Those interactions play a critical role in several processes, including cell proliferation and differentiation, tissue repair (wound healing), fibrosis, and cancer transformation, ultimately driving cells toward distinct phenotypic fates [9,10]. In particular, fibroblast–epithelial interactions modulate tissue repair, homeostasis, inflammation, proliferation, and remodeling, specifically through the mechanotransduction of physical stimuli and the modification of the ECM composition, thus participating in the morphology of epithelial cells and tissues [11].

Under real microgravity conditions, human fibroblast samples cultured during space flight show increased levels of collagen synthesis [12]. Instead, in simulated weightlessness—obtained through specific devices like the Random Positioning Machine (RPM) [13]—fibroblasts undergo several cytoskeleton (CSK) and functional changes, including reduced levels of E-cadherin, α-SMA, and several ECM components [14,15]. Furthermore, a model study of collagen-based three-dimensional matrices showed that fibroblast differentiation and matrix remodeling are impaired in microgravity [16]. Noticeably, the microgravity molds the fibroblast’s shape and orientation, disrupting cell alignment and decreasing a cell’s adherence to the substrate [17]. Therefore, it is tempting to speculate that those effects can impair the fibroblast–keratinocyte interactions and their topographical distribution within the microenvironment, thus altering the overall functioning of the stroma in microgravity conditions.

We, therefore, investigated whether a microgravity (µG) condition affects dermal fibroblast differentiation and their dynamic interaction with collagen and keratinocytes in a co-culture system. Results showed that µG impairs fibroblast motility, invasiveness, and capability in performing an appropriate wound-healing process. Moreover, exposure to simulated weightlessness severely disrupts the physiological architecture on which the fibroblast–keratinocyte interplay relies. This suggests that µG can impair the normal homeostasis of skin tissues.

## 2. Results

### 2.1. Proliferation, Apoptosis, and Cell Cycle

Exposure to µG in the RPM for 24 h induced a statistically significant decrease in cell proliferation in human dermal fibroblasts compared to those grown in normal gravity (*p* < 0.01) (Figure 1A). On the other hand, apoptosis, as determined by a cytofluorimetric assay, significantly increased in human fibroblasts that were exposed to microgravity for 24 h compared to cells grown at 1 g (*p* < 0.05) (Figure 1B). In agreement with proliferation and apoptosis data, fibroblasts subjected to microgravity showed a decrease in the S phase of the cell cycle (*p* < 0.05) and an increase in the G2/M phase (*p* < 0.05) compared to fibroblasts grown under normal gravity conditions (Figure 1C).

### 2.2. Oxidative Stress

An assay based on the detection and quantification of protein carbonylation was used in order to study oxidative stress on human dermal fibroblasts exposed to simulated microgravity. Protein carbonyl derivatives, which represent the most common products of protein oxidation in biological samples, are chemically stable markers of ROS-induced oxidative stress [18]. Exposure of human dermal fibroblasts to µG for 24 h resulted in a statistically significant increase in protein carbonyl derivatives compared with on-ground cells (*p* < 0.05) (Figure 2A,B). In RPM-conditioned fibroblasts, the addition of the antioxidant glutathione (GSH) produced a statistically significant decrease in carbonyl derivatives compared to fibroblasts exposed to µG without the antioxidant (*p* < 0.001) (Figure 2A,B). In order to correlate the increase in oxidative stress in fibroblasts exposed to µG with the apoptotic rate, expression of the pro-apoptotic marker, cl-PARP, was analyzed. As expected, the level of cl-PARP expression was significantly higher in fibroblasts exposed to microgravity for 24 h compared to those cultured on the ground (*p* < 0.01) (Figure 2C,D). The addition of the antioxidant GSH to fibroblasts cultured under microgravity conditions dramatically reduced the expression of cl-PARP (*p* < 0.01 versus RPM) (Figure 2C,D). These findings confirm that an apoptosis surge in fibroblasts exposed to microgravity is closely related to the increase in oxidative stress.

### 2.3. Wound Healing Assay 

In normal conditions, fibroblasts display a highly migrating phenotype, especially when they are challenged by stressful and inflammatory cues. However, exposure to simulated µG seems to impair the migratory capabilities of human dermal fibroblasts, as evidenced by wound repair assay. Indeed, in the wound-healing experiment, after 24 h of fibroblasts culture in RPM, the cell-free area was significantly (*p* < 0.001) greater than that recorded on 1 g control experiment performed (Figure 3A,B). These data showed that motility performances were highly impaired in fibroblasts subjected to microgravity. 

### 2.4. Human Dermal Fibroblasts Migration and Invasion 

Accordingly, a statistically significant decrease in both the migration (*p* < 0.01) and invasion (*p* < 0.001) rate was observed in µG-exposed fibroblasts when compared to on-ground cultured cells (Figure 4A–D). However, a different picture emerged for longer µG exposition times (>72 h), where fibroblasts almost completely recovered their dynamic profile through an increased migratory and invasiveness rate (*p* < 0.05 and *p* < 0.001, respectively, post 24 h of µG). It is worth of noting that a similar result was obtained when µG-conditioned fibroblasts were replaced in on-ground conditions. Those results strongly suggest that the impairment of migrating/invasive properties are adaptive and transient phenotype features emerging in response to the modified balance of biophysical forces within the microenvironment.

### 2.5. Metalloprotease Activity 

The reported reduction in the invasive properties of µG-conditioned fibroblasts is mirrored by observed changes in the metalloproteases (MMPS) activity. The MMP-9 activity decreases after 24 h of the RPM culture with respect to on-ground control (*p* < 0.01) (Figure 5A), while no significant modifications were recorded in MMP2 activity (Figure 5B). Metalloproteases play a key role in wound healing, as well as in cell migration and invasion processes. Therefore, the decrease in MMP-9 is probably instrumental in impairing the migrating/invasive property recorded in µG exposure. 

In order to confirm the invasion capability of the cells, we relied on confocal microscopy to assess the depth of the Matrigel invasion reached by fibroblasts growing in normal gravity or exposed to microgravity for 24 h. As reported in Figure 6, fibroblasts formed scattered clusters in both structures. However, their invasion capability changed significantly. In normal gravity, cells invaded the matrix up to a depth of 320.05 ± 31.36 µm (Figure 6A–C). On the other hand, cells growing in microgravity invaded the matrix up to 145.0 ± 10.40 µm (*p* < 0.01; Figure 6D–F). Overall, such findings confirm that cells exposed to microgravity dramatically reduce their invasiveness capability. 

### 2.6. Molecular Parameters in Human Dermal Fibroblasts Exposed to Simulated Microgravity for 24 h

In human dermal fibroblasts exposed to simulated microgravity for 24 h, we observed a statistically significant decrease in the expression of myofibroblastic markers α-SMA (*p* < 0.01) and Cofilin (*p* < 0.01) as compared to on-ground controls. Furthermore, the exposure of human dermal fibroblasts to simulated microgravity in the RPM for 24 h induced a statistically significant decrease in Fascin expression compared to those grown in normal gravity (*p* < 0.01) (Figure 7A). Noticeably, those changes reverted when fibroblasts were seeded into normal gravity or after longer exposure (>72 h) to µG, thus evidencing that weightlessness-induced modifications should be considered adaptive responses (Figure 7C). WB assay demonstrated a significant reduction in α-SMA quantity, even if the very relevant and unexpected finding is represented by the different distribution of that protein during the microgravity exposure. Indeed, α-SMA translocates from the cytosol to the nucleus after 24 h of µG exposure (Figure 8). This highlighted that a phase transition begins early after cells are seeded into a µG field [19]. It is interesting to note that α-SMA is not present in the nucleus of fibroblasts exposed to RPM for 48 h. Indeed, α-SMA co-localizes with the F-actin fibers along the cell membrane (see Appendix A). In fibroblasts cultured in RPM for 24 h and then transferred into a normal gravitational field for further 24 h (Recovery condition), α-SMA recovers almost completely the distribution pattern observed in 1 g cultured cells (Appendix A).

### 2.7. Cytoskeleton and Cell-Adhesion Changes 

We investigated if µG exposure could impair specific cytoskeleton (CSK) targets involved in enabling cell-surface contact and displacement. Namely, actin-vinculin co-localization is mandatory to allow for cell movement and the transduction of physical forces through the mechanobiological apparatus. As forecasted, confocal microscopy of control and µG-exposed samples revealed a marked disassembly between vinculin and actin filaments at the border of the cell’s membrane and a concomitant decrease in the overall vinculin presence (Figure 9 upper panel). Further, the quantitative analysis of vinculin’s fluorescence intensity (SUM(I)) performed by Leica confocal software confirmed a significant difference between OG and RPM conditions (O.G.: 3324.87 ± 484.03 a.u. versus RPM: 889.50 ± 195.78 a.u; *p* < 0.001). Those findings—together with the impressive reduction of the filopodia/pseudopodia number—further suggest that some key effectors of cell motion are critically hampered during µG exposure. Namely, TEM analysis revealed that cell-to-cell adhesiveness was likely impaired, as suggested by the significant reduction in desmosome-like junctions of µG-exposed cells. Figure 10 shows the ultrastructure of fibroblasts cultured at 1 g (A) or under simulated microgravity (B). Adherent desmosome-like junctions among plasma membrane protrusions were abundant in unitary-gravity-cultured cells, while they were almost completely lacking in simulated-microgravity-exposed samples. Consequently, fibroblasts exposed to weightlessness did not form adhesive junctions, even when cell plasma membranes interacted (Figure 10B). 

It is worth noting that such changes consistently reverted to normalcy after 48 h of µG conditioning. Specifically, the quantitative analysis of the fluorescent intensity of vinculin (SUM(I)) demonstrated that the amount of vinculin present in these cells did not differ significantly from that of cells maintained for 48 h under an OG condition (O.G.: 8615.25 ± 1547.09 a.u. versus RPM: 8553.75 ± 1024.38 a.u; *p* = n.s). This further confirmed that such modifications should be viewed as adaptive and transitory (Figure 9 Lower panel). Indeed, in fibroblasts cultured in RPM for 24 h and then transferred to a normal gravitational field for a further 24 h (recovery condition), vinculin almost completely recovers the distribution pattern observed in 1 g cultured cells (Appendix A).

### 2.8. Collagen Gel Contraction

The reduction of α-SMA and Cofilin in fibroblasts subjected to microgravity is indicative of a de-differentiation from myofibroblasts to quiescent tissue fibroblasts. This phenotypic reversion is confirmed by collagen contraction assay data that showed a statistically significant decrease in the contraction of fibroblasts cultured in the RPM compared to those cultured on the ground (*p* < 0.05) (Figure 11A,B). The fibroblast-induced collagen gel contraction (CGC) assay was established by embedding fibroblasts into a three-dimensional gel matrix, such as collagen, on the bottom of a well plate. This was then manually separated from the well plate surface to loosen the gel puck from the well plate walls and enable contraction [20]. The contractile forces generated by fibroblasts propagated throughout the collagen matrix and arranged collagen fibers to a higher density structure with a decreased matrix volume [21]. Reduction in the size decrease of a gel matrix puck through imaging and subsequent analysis provided a direct way to assess fibroblast contractility. Conversely, an increase in the measured area indicated a reduction in fibroblast-induced contractility.

### 2.9. Co-Culture of Fibroblasts and Keratinocytes

Keratinocytes growing in isolation at 1g showed lower invasive capacity of Matrigel when compared to cells grown in microgravity conditions [22]. Fibroblasts cultured at 1 g invaded the matrix up to a depth of about 390.06 µm, while cells in microgravity invaded it to a depth of about 150 µm (Figure 6). In order to verify the mutual influence on their invasive capability, the two cell types were co-cultured into a thick layer of Matrigel and then exposed to µG. In co-culture experiments, after 24 h of conditioning in normal gravity, fibroblasts and keratinocytes invaded the Matrigel and formed three-dimensional, regular, ovoid structures, in which cells were randomly distributed (Figure 12A,C). On the contrary, fibroblasts and keratinocytes co-cultured in µG for 24 h, after Matrigel invasion, gave rise to branched-out organoid-like structures, in which the two cell types were partitioned into distinct cell layers, with the keratinocytes in the upper part of the aggregate (Figure 12B,D). 

## 3. Discussion

Gravity is a fundamental physical constraint that dramatically affects the structure and the behavior of living organisms. This has been highlighted by an impressive body of evidence gathered during the last 50 years, since the beginning of human space exploration [23]. Noticeably, even mild reductions in the gravitational force can trigger the remodeling of living functions at different levels, from the macroscale (body fluids, organs, and tissues) to the nanoscale (dynamics and structure of proteins and other molecular components). Cells exposed to microgravity are profoundly affected by the physical changes occurring in this unique environment, which include loss of gravity-dependent convection, disappearance of hydrodynamic shear [24], and lack of sedimentation. Moreover, several pathways are sensitive to even mild changes in the balance between those forces that affect the morphogenetic field. According to non-equilibrium thermodynamics that govern dissipative processes, these can display unexpected behaviors, such as cytoskeleton arrangement, cell division, and mechanotransduction of physical stresses, to name a few [25,26]. Specifically, dynamic adhesion processes between cells and their microenvironment are profoundly affected in µG, leading to bewildering modification in cellular aggregation and tissue architecture [27,28]. 

Fibroblasts play a critical role in tissue homeostasis because they behave as true “transducers” of many interactions between cells and their microenvironment. Fibroblasts are active players during wound-healing processes [29] and inflammatory reactions [30]. In fact, they convey several biochemical and physical cues that ultimately modify the response of the epithelial layer. Noticeably, under the stimulus of inflammatory (e.g., TGF-β release) or physical (e.g., increased stiffness) stresses, fibroblasts differentiate into myofibroblasts [31]. The increased expression of alpha smooth muscle actin (α-SMA) is instrumental in supporting the transition from fibroblasts to myofibroblasts, specifically by allowing myofibroblasts to gain contractile properties and physically remodel ECM structure and composition [32].

Only a few studies have investigated the performance of fibroblasts in µG conditions. Fibroblasts growing in weightlessness exhibit significant changes in collagen synthesis, cytoskeleton architecture, and ECM production (e.g., laminin and fibronectin), as well as altered growth behavior [12]. It is worth noting that in microgravity, several genes were either down-regulated or up-regulated [33], while the expression of both E-cadherin and α-SMA genes were found to be significantly reduced [34]. 

Here, we show that fibroblasts cultured in microgravity underwent several changes that impaired their differentiation into myofibroblasts. The cell cycle after 24 h of µG conditioning showed slight, albeit significant changes with an increased apoptosis rate and reduced proliferation. During this period, fibroblasts experienced significant oxidative damage (Figure 2) that was successfully rescued when GSH was added to cell cultures. Microgravity-conditioned fibroblasts were unable to mount an efficient wound-healing response (Figure 3) as their migrating properties were severely compromised (Figure 4). This occurred alongside a reduced capability to degrade the surrounding microenvironment due to the diminished release of MMP (Figure 5), both in 2D as well as 3D cell culture models (Figure 6). Those changes were supported by an impressive down-regulation of those molecular parameters (Fascin, α-SMA, Cofilin) and cytoskeleton-related effectors (actin and vinculin), which participate in modulating cell motility. It is worth noting that a loss of vinculin significantly impairs traction force generation and motility [35]. However, this effect can be appreciated only in 3D matrices [36] (or in µG conditions), given that fibroblasts lacking vinculin displayed a paradoxical increase in migration rates in 2D cultures [37]. Despite cell-to-cell adhesiveness being weakened, as documented by the significant lessening of desmosome-like junctions, the overall migrating capability of µG-exposed fibroblasts was impaired by the concomitant down-regulation of α-SMA. Over-expression of α-SMA is required to facilitate motility and contractility. Particularly, wound contraction takes place when a *de novo* expressed α-SMA is incorporated in *stress fibers* [38]. In turn, the α-SMA mediated contraction plays a critical role in mechanically regulating fibroblast fate by controlling YAP/TAZ activation and, ultimately, modulating the overall transduction of physical forces [39]. Therefore, the reduced α-SMA expression can explain the significant impairment in the collagen gel contraction that we observed with µG-conditioned fibroblasts (Figure 10). Noticeably, changes in migration and invasiveness capability were transitory, as they reverted after 72 h of µG exposure or when fibroblasts were seeded again in a normal gravity field. Yet, the observed displacement of α-SMA within the nucleoplasm represented a mark of cell phenotypic transition, and this finding could explain how non-genetic modifications can be instrumental in driving a system toward different cell states of differentiation [16,19].

Indeed, we recorded some modifications in gene expression during an exposure to weightlessness. Nevertheless, it is unlikely that the impressive phenotypic changes observed in fibroblasts can be ascribed to any genome activity fine-tuning. In fact, as previously reported, RNA sequencing has revealed minimal variation in transcriptomic of fibroblasts under µG conditions [13,16], while phenotypic changes in µG are most likely driven by non-genetic mechanisms acting at the system biology level and governed according to the non-equilibrium thermodynamics principles [26]. 

Amazingly, as per our previous study [22], the fate of keratinocytes in microgravity showed an opposite trend, given that they acquired several traits pertaining to the EMT phenotype. Keratinocytes show an increased expression of EMT-related markers and the development of organoid-like structures, while losing cell-to-cell adhesions. This latter aspect is instrumental in enabling the disassembly of cell clusters and in promoting their migratory capability. Therefore, changes occurring in the motility properties of both keratinocytes and fibroblasts exposed to µG may have favored the establishment of new reciprocal relationships between cell types. Overall, these changes were instrumental in reframing the interactions among different cells and, likely, between cells and their microenvironment. This is precisely what we observed when both keratinocytes and fibroblasts were co-cultured all together in µG.

We showed that µG significantly modified the way fibroblasts and keratinocytes interacted in reshaping the 3D architecture of the organoid that resulted from cell co-culture within the Matrigel. When the co-culture was exposed to µG, the normal fibroblasts–keratinocytes interplay was disrupted, and the two cell types were partitioned into distinct cell layers, with the keratinocytes in the internal core of the aggregate. On the contrary, under normal gravity conditions, fibroblasts and keratinocytes invaded the Matrigel and formed three-dimensional, regular ovoid structures in which cells were randomly distributed. This is a relevant outcome, as it suggests that µG may significantly alter the normal tissue organization, challenging the physiological function of the system, namely by modifying the way a living system responds to even physiological inputs [40]. Specifically, these findings demonstrated that modifications in the surrounding microenvironment, which involve a rewiring of critical constraints (like gravity), are critical in shaping cells and tissues of three-dimensional structures.

Those modifications are probably adaptive, as observed from the many changes occurring during the early period of µG exposure. However, further studies on prolonged periods of µG exposure are needed to assess if µG could represent a critical threat for living beings and the functioning of complex tissues, as well investigate the molecular pathways involved in the observed changes. Given that µG exerts distinct effects on the adhesiveness of keratinocytes and fibroblasts, it is tempting to speculate that this would influence the transduction of mechanical stimuli from the microenvironment. Impaired mechanotransduction can likely affect the crosstalk among the two cell types, their interaction with ECM, and the production of soluble factors, which, in turn, can control the reciprocal behavior of the two cell clusters [41].

## 4. Materials and Methods

### 4.1. Cell Culture

Primary cultures of human dermal fibroblasts were obtained by a collagenase (625 U/mL; Sigma-Aldrich, Merck KGaA, Darmstadt, Germany) digestion as previously described [42] and cultured in a Dulbecco Modified Eagle Medium (DMEM) supplemented with 10% Fetal Bovine Serum (FBS) and antibiotics (penicillin 100 IU/mL, streptomycin 100 μg/mL, all from Sigma-Aldrich). The cells were cultured at 37 °C in an atmosphere of 5% CO2 in the air. A skin biopsy was obtained from female healthy donors (*n* = 5) who underwent a reductive mastectomy. The cells from different donors were pooled all together. Experiments in duplicate/triplicate were performed using this same pool. The use of clinical samples complied with the Declaration of Helsinki 1975, revised in 2008, and was approved by the Ethical Committee of Sapienza University of Rome (Ref. 3778-prot. 2464/15). Written consent was obtained before inclusion in the study.

The human keratinocytes cell line HaCaT (kindly provided by Prof. M. R. Torrisi, Sapienza University, Rome, Italy) was cultured in a Dulbecco Modified Eagle Medium (DMEM) supplemented with 10% Fetal Bovine Serum (FBS) and antibiotics (penicillin 100 IU/mL, streptomycin 100 µg/mL, all from Euroclone Ltd., Cramlington, UK). The cells were cultured at 37 °C in an atmosphere of 5% CO2 in the air.

Co-culture human dermal fibroblasts-human keratinocytes cell line HaCaT was used. Primary cultures of human dermal fibroblasts and human keratinocytes cell line HaCaT was cultured on a plastic dish and on GFR Matrigel. Fibroblasts were labeled using the PKH-67 Green Fluorescent Cell Linker Kit (Sigma-Aldrich, St. Louis, MO, USA), and human keratinocytes were labeled using the PKH-26 Red Fluorescent Cell Linker Kit according to the manufacturer’s instructions.

### 4.2. Simulated Microgravity and Cell Exposure

Microgravity conditions were simulated by a desktop RPM, which is a particular kind of 3D clinostat [22] manufactured by Dutch Space (Leiden, The Netherlands). The desktop RPM was positioned in a standard incubator. The RPM was set at an angular velocity of 60/s. Human dermal fibroblasts were seeded in 25 cm^2^ flasks (T25, Falcon, Becton–Dickinson Labware, FranklinLakes, NJ, USA) and were used for experiments. Before exposing the cells to the regime of microgravity, the flasks were filled completely with fresh culture medium to eliminate the presence of air bubbles and therefore decrease the effects of turbulence and shear stress during rotation. Flasks containing fibroblast sub-confluent monolayers were fixed into the RPM as close as possible to the center of the platform. The RPM and on-ground (OG) cultures were kept in the same humidified incubator at 37 °C in an atmosphere of 5% CO2. Fibroblasts were placed in the RPM for 24 h to investigate the early changes in the functional and morphological phenotype. To assess the adaptive process to microgravity, the fibroblasts were placed in the RPM for 72 h. For recovery experiments, the fibroblasts were placed in the RPM for 24 h and then transferred into a normal gravitational field for an additional 24 h.

Co-culture fibroblasts-keratinocytes, labeled in the cell culture paragraph as described, were seeded in 25 cm^2^ flasks and used for experiments. After reaching the sub-confluence, the flasks, filled completely with fresh medium, were fixed into the RPM. The RPM and OG co-cultures were kept in a humidified incubator at 37 °C in an atmosphere of 5% CO2 in the air. Co-cultures were placed in the RPM and the OG for 24 h on GFR Matrigel.

### 4.3. Human Dermal Fibroblast Proliferation, Apoptosis, and Cell Cycle

For the proliferation assay, human dermal fibroblasts were treated for 24 h in the RPM or kept on the ground, before being trypsinized and counted with a particle count and size analyzer (Beckman-Coulter, Inc., Fullerton, CA, USA). For the apoptosis assay, human dermal fibroblasts, exposed or unexposed to simulated microgravity for 24 h, were trypsinized and washed twice with PBS. The cells were stained with FITC-labeled Annexin V/7-AAD (7-aminoactinomycin- D) according to the manufacturer’s instructions (Annexin V/7-AAD kit; Immunotech, Marseille, France). Briefly, a washed cell pellet was re-suspended in a 500 μL binding buffer at a concentration of 106 cells/mL; 10 μL of annexin V, together with 20 μL 7-AAD, was added to 470 μL cell suspension. The cells were incubated for 15 min on ice in the dark. The samples were analyzed by flow cytometry. For the cell cycle assay, human dermal fibroblasts were incubated for 24 h under normal gravity (OG) and in simulated microgravity conditions. Fibroblasts were then harvested, washed twice with PBS, fixed with 70% ethanol at 4 °C for 24 h, and stained with an aqueous staining solution containing 0.75 mM (0.5 mg/mL) Propidium Iodide (PI), 4 mM sodium citrate, 1% Triton-X100, and 1% bovine serum albumin (all from Sigma Chemical Co.) at 4 °C overnight. PI-stained cells were measured by flow cytometry. For each type of assay, three independent experiments were performed in triplicate.

### 4.4. Detection of Protein Carbonylation by Western Blotting

To analyze the protein carbonyl content in lysates of human dermal fibroblasts exposed or unexposed to simulated microgravity for 24 h, pre-boiled protein samples (15 μg of total protein/lane) were subjected to 10% SDS-PAGE for protein separation and transferred to the nitrocellulose membrane. Protein carbonyls were detected using the immunological method OxySelect protein carbonyl immunoblot kit (Cell Biolabs, San Diego, CA, USA) according to the manufacturer’s instructions. Briefly, proteins transferred to the nitrocellulose membrane were derivatized with DNPH and probed with rabbit anti-DNP antibody. Antigen-antibody complexes were visualized by the ECL reagent and quantified spectrophotometrically. In order to confirm the correlation between protein carbonylation and oxidative stress, fibroblasts, cultivated in the RPM or not, were treated with the antioxidant glutathione (GSH) (5 mM) (Sigma-Aldrich).

### 4.5. Wound-Healing Assay

The wound-healing assay was performed using special double-well culture inserts (ibidi GmbH, Martinsried, Germany). Each insert was placed in a well of an 8-well μ-slide (ibidi), and 3.5 × 10^4^ human dermal fibroblasts were placed into both wells of each insert, with 70 μL of medium containing 10% FBS. At confluence, the culture inserts were gently removed, and fibroblasts were fed with fresh medium. Each well was photographed at 10x magnification immediately after the insert removal for baseline wound measurement (T0). The slides were then exposed to simulated microgravity condition (RPM) or kept on the ground as control (OG) for 24 h, where each well was once again photographed at 10x magnification. Photos were taken with a Nikon DS-Fi1 camera (Nikon Corporation, Tokyo, Japan) coupled with a Zeiss Axiovert optical microscope (Zeiss, Oberkochen, Germany). The mean percentage of the residual open area compared to the respective cell-free surface in T0 was calculated using ImageJ v 1.47 h software. For each experimental condition, four independent experiments were performed.

### 4.6. MMPs Gelatin Zymography

The enzymatic activities of MMP2 and MMP9 were determined by gelatin zymography. Briefly, the same amount of conditioned media of human dermal fibroblasts, exposed or unexposed to simulated microgravity for 24 h, was prepared with a standard SDS-polyacrylamide gel loading buffer containing 0.01% SDS without β-mercaptoethanol and not boiled before loading. Then, prepared samples were subjected to electrophoresis with a 12% SDS–PAGE containing 1% gelatin. After electrophoresis, gels were washed twice with distilled water containing 2.5% Triton X-100 for 30 min at 25 °C to remove SDS. Gels were then incubated in a collagenase buffer (0.5 M Tris-HCl pH 7.5, 50 mM CaCl2 and 2 M NaCl) overnight at 37 °C, stained with Coomassie brilliant blue R-250, and destained with destaining solution (30% methanol, 10% acetic acid, and 60% water). Three independent experiments were performed.

### 4.7. Preparation of Cellular Extracts and Western Blot Analysis

Human dermal fibroblasts, either exposed to simulated microgravity in the RPM or cultured in normal gravity (1 g), were washed twice with ice-cold PBS and scraped in RIPA lysis buffer (Sigma-Aldrich). A mix of protease inhibitors (Complete-Mini Protease Inhibitor Cocktail Tablets, Roche, Mannheim, Germany) and phosphatase inhibitors (PhosStop; Roche) was added just before use. Cellular extracts were then centrifuged at 8000 g for 10 min. The protein content of supernatants was determined using the Bradford assay. For Western blot analysis, an equal amount of protein extracts was loaded into each well and separated on Mini PROTEAN TGX precast gels (BIO-RAD, Bio-Rad Laboratories, Hercules, CA, USA). Proteins were blotted onto nitrocellulose membranes (BIO-RAD) and probed with the following antibodies: polyclonal anti-α-SMA (ab5694, from Abcam, Cambridge, UK); polyclonal anti-Fascin 1 (sc-28265, from Santa Cruz Biotechnology) and Cofilin (sc-33779, from Santa Cruz Biotechnology); monoclonal anti-β-actin (A2547, Sigma-Aldrich); and polyclonal anti-cleaved PARP (#9541) and rabbit monoclonal anti-GAPDH (#2118), both from Cell Signaling Technology. Then, the membranes were incubated with the appropriate HRP-conjugated secondary antibody (GE Healthcare, Little Chalfont, UK) for 1 h at room temperature. Immunocomplexes were detected with an enhanced chemiluminescence kit (WesternBright ECL HRP Substrate, Advansta Inc., Menlo Park, CA, USA) according to the manufacturer’s instructions. All Western blot images were acquired and analyzed through an Imaging Fluor S densitometer (Bio-rad). Each experiment was performed three times.

### 4.8. Collagen Gel Contraction

After 24 h of exposure to simulated microgravity (RPM) or on-ground culture, human dermal fibroblasts were suspended in a collagen solution (rat tail collagen type I, BD Biosciences, San Jose, CA, USA) at a concentration of 0.75 mg/mL and seeded into the wells of a 24-well plate. After collagen polymerization, the gels were gently detached from the wells, and the complete culture medium (DMEM 10% FCS) was added. The plates were incubated at 37 °C in a 5% CO2 atmosphere for 18 h. The contraction of each sample was photographed, and the diameter of each collagen gel was measured using ImageJ v 1.47 h software. Each experiment was performed three times in duplicate.

### 4.9. Cell Migration and Invasion Assays

Human dermal fibroblasts, either exposed to simulated microgravity in the RPM or cultured in normal gravity (1 g), were placed (5 × 10^3^ cells) in DMEM 0.1% FBS in the upper side of 8-μm filters (Falcon, BD Biosciences) for the migration assay or in Matrigel-coated 8 μm filters (BD Bio-Coat™ growth factor reduced MATRIGEL™ invasion chamber, Falcon, BD Biosciences) for the invasion assay. They were then placed in the wells of a 24-well plate (Falcon, BD Biosciences), containing 0.8 mL of DMEM 10% FBS. After 24 h of incubation at 37 °C, cells from the upper surface of the filters were removed with gentle swabbing, and cells that had migrated to or invaded across the Matrigel’s lower surface were fixed and stained with hematoxylin/eosin. The membranes were examined microscopically to determine cellular migration and invasion by cell counting in at least 4–5 randomly selected fields for each. A Zeiss Axiovert 10 optical microscope was used. For each data point, four independent experiments in duplicate were performed.

### 4.10. Transmission Electron Microscopy (TEM) Analysis

Fibroblasts cultured at unitary gravity or under simulated microgravity were fixed in 2.5% glutaraldehyde in 0.1 M cacodylate buffer (pH 7.4), postfixed in 1% OsO4, and treated with 1% tannic acid. The cells were then gently mechanically detached maintaining the cell-to-cell contacts, suspended in cacodylate buffer, de-hydrated in ethanol, and embedded in epoxy resin. Ultrathin sections were contrasted in aqueous lead-hydroxide, followed by tannic acid treatment, and photographed by a Libra 120 Transmission Electron Microscope (TEM) equipped with a wide-angle dual-speed CCD camera Sharpeye 2K(4Mpx) operated by iTEM software (Soft Image System, Münster, Germany).

### 4.11. Immunofluorescence and Confocal Analysis

Fluorescent analysis was performed in cells grown until semi-confluence. Fibroblast cells were cultured into 8-well µ-slides (ibidi GmbH, AmKlopferspitz19, D-82152 Martinsried, Germany) and exposed or unexposed to simulated microgravity for 24 h. At the end of incubation, the cells were fixed with 4% paraformaldehyde for 10 min at 4 °C and washed twice for 10 min with PBS. The cells were permeabilized for 30 min using PBS, 3% BSA (Bovine Serum Albumin, Santa Cruz Biotechnology), and 0.1% Triton X-100 (Sigma-Aldrich), followed by mouse anti-vinculin (Santa Cruz, Cat sc- 73614 1:50 dilution) and by rabbit anti-α-SMA (Abcam, Cat ab5694 1:100 dilution).

The cells were washed with PBS and incubated for 1 h at room temperature with the opportune secondary antibody (FITC-AffiniPure donkey anti-mouse IgG Cat# 715-095-150, RRID: AB_2340792, purchased from Jackson Immuno Research Labs) and rhodamine phalloidin (Invitrogen Molecular Probes Eugene 1:40 dilution) for F-actin visualization. The slides were then washed with PBS and mounted with 0.1 mM Tris-HCl at pH 9.5: glycerol (2:3). Negative controls were processed in the same conditions besides primary antibody staining. Finally, immune localization was analyzed with a Leica confocal microscope TCS SP2 (Leica Microsystems Heidelberg GmbH, Mannheim, Germany) equipped with Ar/ArKr and He/Ne lasers. The images were scanned under a 20× lens.

Analysis of GFR Matrigel invasion was run on fibroblasts alone and on a co-culture of fibroblasts and keratinocytes, which were labeled as stated in the Cell Culture section. For depth analyses, optical spatial series with a step size of 2 µm were recovered. All analyses were performed using Leica Confocal software. Data were analyzed with Sigma Plot software.

## 5. Conclusions

Microgravity impairs tissue organization and critical pathways associated with several repair-related processes, including wound healing. Both tissue architecture and complex physiological functions depend on proper cooperation between cells and their microenvironment. In this setting, a critical role is supported by fibroblasts that undergo a number of cell fate transitions to acquire distinctive differentiated capabilities. Our findings show that fibroblast differentiation is severely impaired after µG exposure for short periods. Fibroblasts’ conversion into myofibroblasts is inhibited alongside their migratory and invasive properties. Consequently, the normal interplay between fibroblasts and keratinocytes in 3D co-culture experiments was remarkably altered, giving rise to abnormalities in organoid-like structures. The downregulation of α-SMA and its unexpected translocation in the nucleoplasm, which is associated with the parallel modification of the actin-vinculin apparatus, can explain why fibroblast-related contractility and mechanotransduction are remarkably modified during µG exposure. These phenotypic changes can likely be triggered by the oxidative damage enacted by the stress associated with weightlessness. There are further ongoing studies to investigate if an appropriate countermeasure—including antioxidant strategies—could help in damping microgravity-induced disruptive effects on fibroblasts and tissue organization

## Figures and Tables

**Figure 1 ijms-23-02163-f001:**
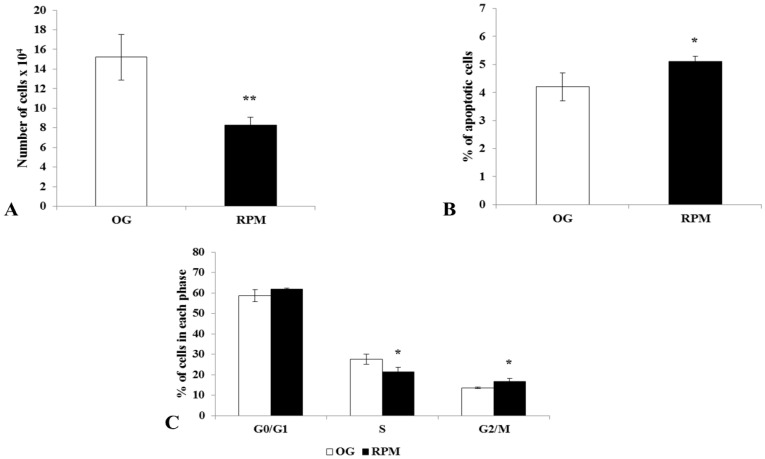
Proliferation (**A**), apoptosis (**B**), and the cell cycle (**C**) in human dermal fibroblasts exposed to simulated microgravity (RPM) for 24 h. (**A**) Cell proliferation was determined by cell count assays performed by a particle count and size analyzer. (**B**) Apoptosis was determined by Annexin V-FITC and 7-AAD staining flow cytometry; histogram shows the percentage of apoptotic cells. (**C**) Cell cycle distribution was determined by propidium iodide staining flow cytometry. The histogram shows the percentage of cells in various phases of the cell cycle. Results are the mean ± SD of three independent experiments performed in triplicate. * *p* < 0.05, ** *p* < 0.01 RPM versus on-ground control (OG) by unpaired two-tailed *t*-tests.

**Figure 2 ijms-23-02163-f002:**
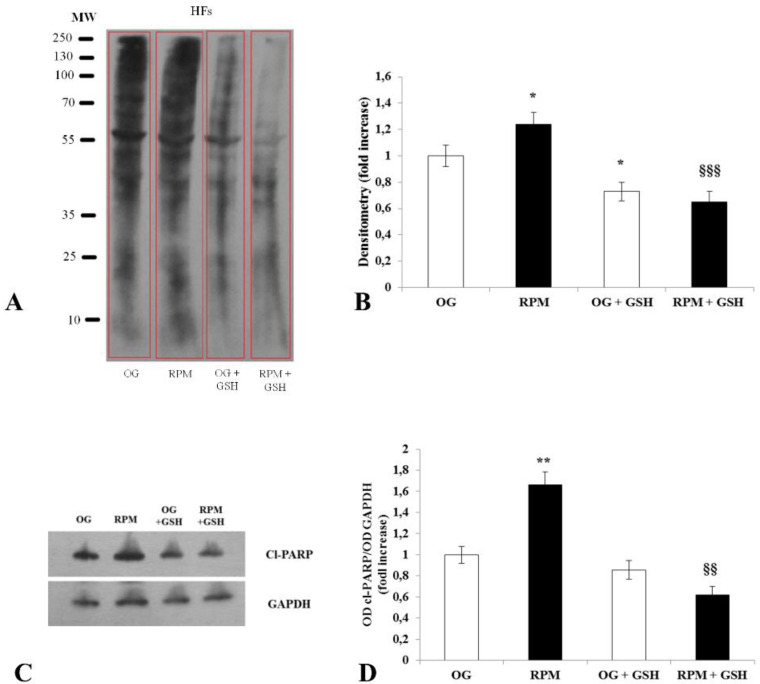
Oxidative stress in human dermal fibroblasts after being exposed to simulated microgravity for 24 h. (**A**) Immunoblots for total carbonylated proteins from human fibroblasts exposed or unexposed to the RPM, treated or not with the antioxidant glutathione GSH. (**B**) Band densitometry analysis for total carbonylated proteins. Values are expressed as a fold increase of OG control, where each column represents the mean value ± SD of four independent experiments. SD is depicted as vertical bars. * *p* < 0.05 versus OG control; ^§§§^
*p* < 0.001 versus RPM by ANOVA followed by Bonferroni post-test. (**C**) Representative Western blot analysis panel and (**D**) immunoblot bar charts showing the expression of Cl-PARP in human dermal fibroblasts exposed or unexposed to simulated microgravity, treated or untreated by GSH. Columns represent the densitometric quantification of optical density (OD) of a Cl-PARP signal normalized with the OD values of GAPHD as a loading control. Values are expressed as a fold increase of OG control. Each column represents the mean value ± SD of three independent experiments, where SD is depicted as vertical bars. ** *p* < 0.01 versus OG control; ^§§^
*p* < 0.01 versus RPM by ANOVA followed by Bonferroni post-test.

**Figure 3 ijms-23-02163-f003:**
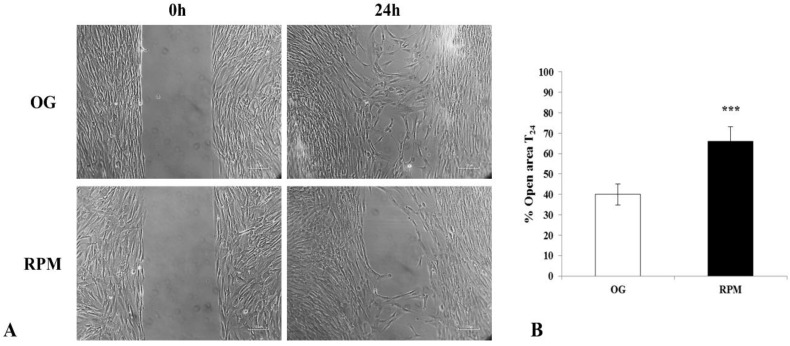
Effects of simulated microgravity on wound healing in human dermal fibroblasts at 24 h. (**A**) Representative images show human fibroblasts exposed or unexposed to the RPM for 24 h at 0 and 24 h. Images were obtained by optical microscopy with 100× magnification. (**B**) Values are expressed as the percentage of open area, measured using ImageJ v 1.47 h software; the value of the open area at 0 h is 100%. Each column represents the mean value ± SD of four independent experiments with standard deviations represented by vertical bars. *** *p* < 0.001 RPM versus on-ground control (OG) by unpaired two-tailed *t*-test.

**Figure 4 ijms-23-02163-f004:**
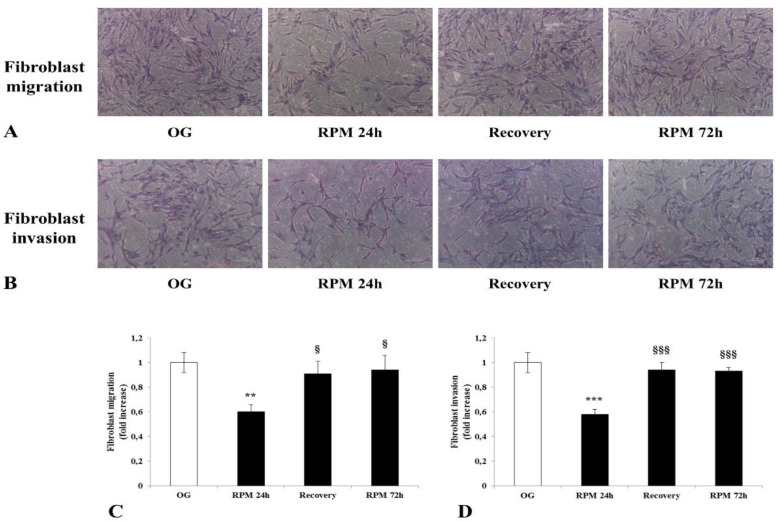
Effect of simulated microgravity on the migration (**A**,**C**) and the invasion (**B**,**D**) of human dermal fibroblasts. Transwell migration and invasion assays were performed on human fibroblasts exposed to the RPM for 24 h, 72 h, and after recovery experiments. Images were obtained by optical microscopy with 100× magnification. Values are expressed as a fold increase of on-ground control (OG). Each column represents the mean value ± SD of four independent experiments performed in duplicate with SD represented by vertical bars. ** *p* < 0.01; *** *p* < 0.001 versus OG; ^§^
*p* < 0.05; ^§§§^
*p* < 0.001 versus RPM 24 h by ANOVA followed by Bonferroni post-test.

**Figure 5 ijms-23-02163-f005:**
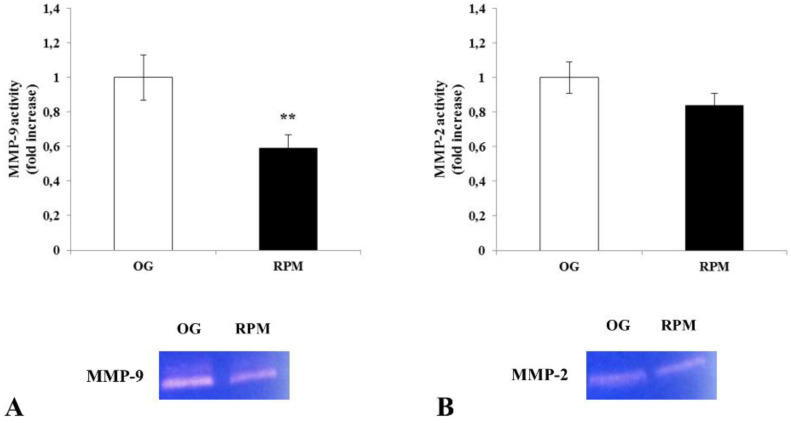
Effect of simulated microgravity on MMP-9 (**A**) and MMP-2 (**B**) activity in human dermal fibroblasts. Gelatin zymography shows MMP-9 and MMP-2 activity in conditioned media of human fibroblasts exposed or unexposed to the RPM for 24 h. Values are expressed as a fold increase of on-ground control (OG). Each column represents the mean value ± SD of three independent experiments, where SD is depicted as vertical bars. ** *p* < 0.01 RPM versus OG by unpaired two-tailed *t*-test.

**Figure 6 ijms-23-02163-f006:**
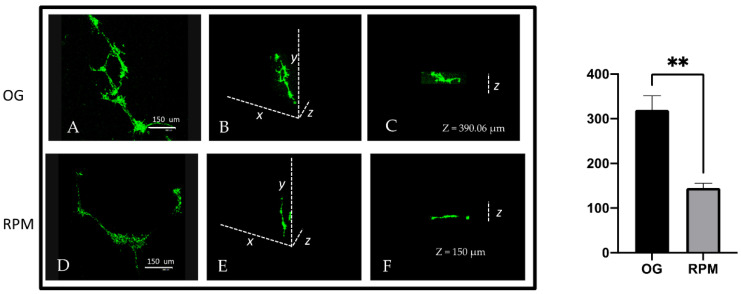
Cells (green) were cultured on GFR Matrigel under normal gravitational (**A**–**C**) and microgravity conditions (**D**–**F**) for 24 h. A, D represent the confocal maximum projection. B, E represent 3D projection. C, F show the depth analysis. As shown in this figure, cells maintained at unit gravity invaded the matrix to a depth of about 390.06 µm (C), while cells maintained under a microgravity condition were able to invade to a depth of about 150 µm (F). The graph indicates the mean value ± SE of depth along the zeta axis measured at a 4-spatial series. (** 1 g versus RPM *p* < 0.01; scale bar 150 µm).

**Figure 7 ijms-23-02163-f007:**
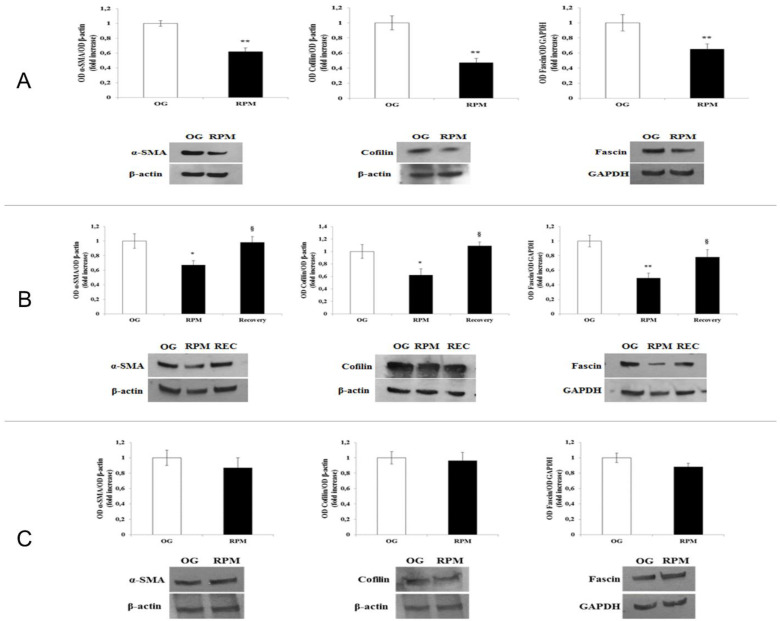
Western blot analyses of α-SMA, Cofilin, and Fascin in human dermal fibroblasts exposed to simulated microgravity (RPM). Columns represent densitometric quantification of the optical density (OD) of a specific protein signal normalized with the OD values of the β-actin (α-SMA and Cofilin) and GAPHD (Fascin). Values are expressed as a fold increase of on-ground control (OG). Each column represents the mean value ± SD of three independent experiments, where SD is depicted as vertical bars. (**A**) Fibroblasts exposed to simulated weightlessness (RPM) and in normal gravity (OG) for 24 h. Each column represents the mean value ± SD of three independent experiments, where SD is depicted as vertical bars. ** *p* < 0.01 RPM versus OG by an unpaired two-tailed *t*-test. On the bottom, a panel of representative Western blot analyses is reported. (**B**) Same conditions as in A: cells are reseeded into normal gravity (recovery) for 24 h. ** *p* < 0.01 RPM versus OG; ^§^
*p* < 0.05 recovery versus RPM by ANOVA followed by Bonferroni post-test. (**C**). Western blot analyses of α-SMA, Cofilin and Fascin in human dermal fibroblasts exposed to simulated microgravity (RPM) for 72 h. Data were statistically analyzed with unpaired two-tailed *t*-test. No significant differences were recorded between on-ground and in-microgravity cultured cells after 72 h of µG. On the bottom, a panel of representative Western blot analyses is reported.

**Figure 8 ijms-23-02163-f008:**
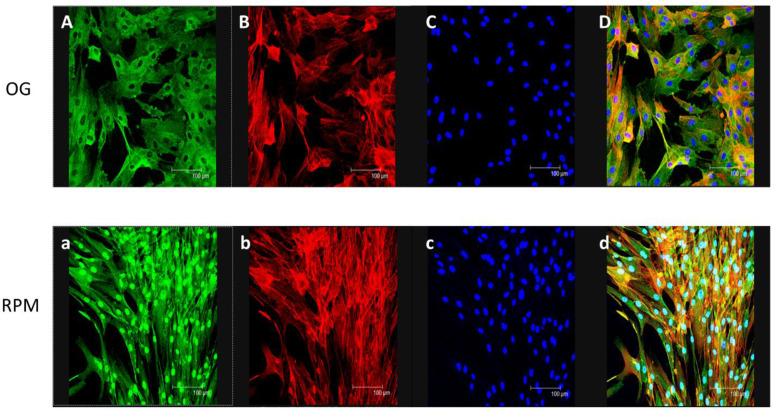
Representative confocal maximum projection of α-SMA (green) and F-actin (red) in fibroblasts cultured either in normal gravity (OG) or in µG (RPM) for 24 h. distribution appears reduced in cytoplasm of fibroblasts cultured in µG (**a**) when compared to cells growing in 1 g (**A**). In (**B**,**b**) F-actin distribution is shown. (**C**,**c**) represent TO-PRO-3 staining for nuclei. (**D**,**d**) represent the merge images. α-SMA localizes within the nucleus, as evident by the picture (**d**) (scale bar 100 µm).

**Figure 9 ijms-23-02163-f009:**
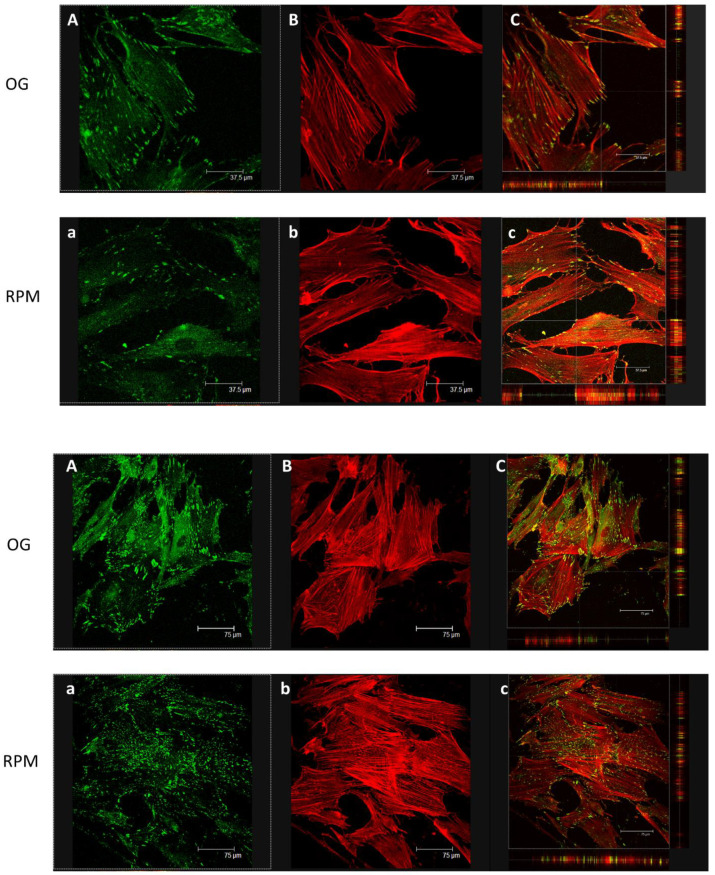
(**Upper panel**) Representative confocal images of vinculin (green) and actin (red) in cells cultured under normal gravitational conditions (OG) and microgravity conditions (RPM) for 24 h. Fibroblasts in µG showed a remarkable reduction in vinculin distribution (**a**), particularly behind the cytoplasmic membrane and close to the cell’s free front, as well as a reduction of stress fibers (**b**) compared to OG condition (**B**). Vinculin (green) and actin (red) co-localized in fibroblasts cultured in normal gravity (**C**), especially at the level of protrusive structures (filopodia and pseudopodia). Little or no co-localization was observed at the level of membrane protrusion in an RPM-cultured cell (**c**) (scale bar 37.5 µm). (**Lower panel**) Representative confocal images of vinculin (green) and F-actin (red) in cells cultured in normal gravitational condition (OG) and microgravity conditions (RPM) for 48 h. Under RPM conditions, the organization of stress fibers was similar to that observed in cells maintained in normal gravitational (OG) conditions. (**C**,**c**) show Vinculin (green) and actin (red) co-localization (scale bar 75 µm).

**Figure 10 ijms-23-02163-f010:**
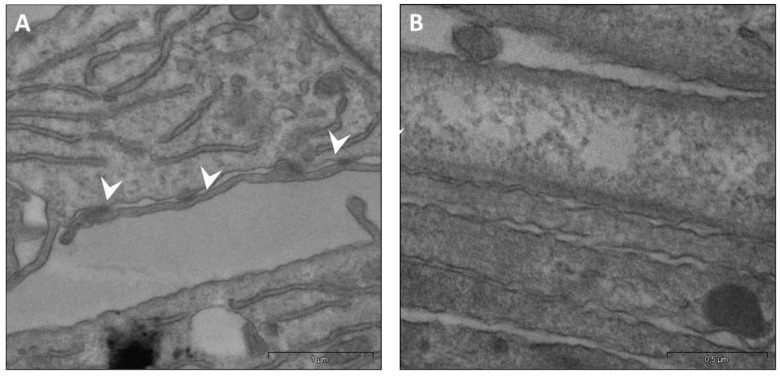
TEM analysis of fibroblasts cultured at unitary gravity or under simulated microgravity for 24 h. The ultrastructure of fibroblasts cultured at unitary gravity (**A**) or under simulated microgravity (**B**) is shown. The arrowheads indicate adherent desmosome-like junctions among plasma membrane protrusions, which were abundant in unitary-gravity-cultured cells (**A**) but could not be observed in samples exposed to simulated microgravity. Indeed, in the latter samples, cells did not form adhesive junctions, even when the cell plasma membranes interacted (**B**).

**Figure 11 ijms-23-02163-f011:**
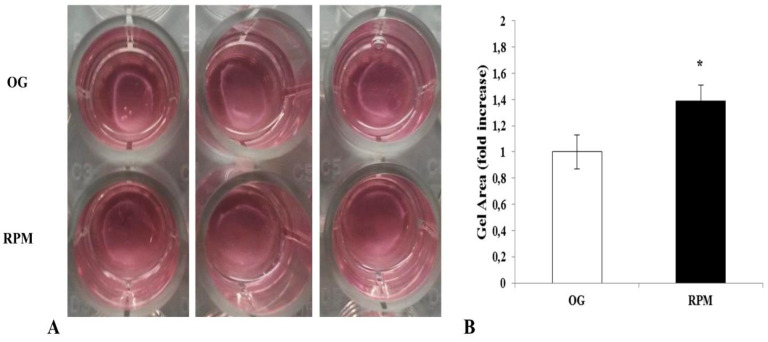
Effects of simulated microgravity on collagen gel matrix contraction of human dermal fibroblasts exposed or unexposed to the RPM for 24 h. The diameter of each collagen gel was measured using ImageJ v 1.47 h software. (**A**) Photos show the action of human fibroblasts exposed to µG (RPM) or to 1g (OG) on collagen gel contraction. (**B**) Values are expressed as a fold increase of on-ground control (OG). Each column represents the mean value ± SD of three independent experiments performed in duplicate, with SD represented by vertical bars. * *p* < 0.05 versus OG by an unpaired two-tailed *t*-test.

**Figure 12 ijms-23-02163-f012:**
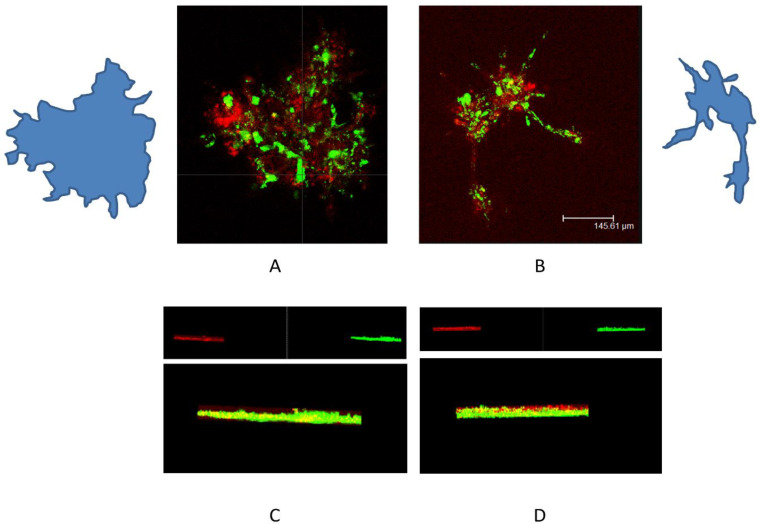
Representative confocal images of fibroblast (green) and keratinocytes (red) co-cultured on Matrigel in normal gravitational (**A**) and microgravity conditions (**B**) for 24 h. (**C**,**D**) show the three-dimensional Leica software reconstructions. The confocal microscope analysis showed that in OG, the fibroblasts were arranged randomly with keratinocytes, while in the RPM, the fibroblasts and keratinocytes had distinct positions (scale bar 145.6 µm).

## Data Availability

Not applicable.

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
