# Peer review of "Microgravity Modifies the Phenotype of Fibroblast and Promotes Remodeling of the Fibroblast–Keratinocyte Interaction in a 3D Co-Culture Model"

_ijms, 2022, doi:10.3390/ijms23042163_

Round 1

Reviewer 1 Report

Authors have investigated the effect of simulated microgravity on the behavior of fibroblasts cultured in a 3D Matrigel system. A number of different parameters have been investigated, although most of findings have been already reported in previous literature.

Interestingly, the interactions between fibroblasts and keratinocytes were also explored.

Comments:

  • Authors report results after 24 or 48 or72 h. Since a number of changes have been considered an adaptive response, with differences observed at 24 being cancelled at 48/72h, it will be important to have all parameters measured for the same length of time (72h), in order to clarify if all or only some parameters are permanently or temporarily modified.
  • All experiments were performed with primary cultures of fibroblasts isolated from one patient. Since behavior of cells in primary cell culture is highly variable, results from only one cell culture should be considered preliminary and discussion should be modified accordingly.
  • The co-culture system represents a rather innovative approach in the context of microgravity-experiments, however this aspect was evaluated only in terms of morphological observations and therefore it appears disconnected with other results. Apart from the different organization of the two cell types, it would be of interest if these changes modify fibroblasts’ behavior, however, all other experiments were performed with fibroblasts alone. Authors should explain the rationale of the experimental design.
  • Paragraph on wound healing assay: Authors state that in normal conditions fibroblasts exhibit an invasive phenotype. Since the “invasive phenotype” is frequently associated to cancer cells, I would suggest using a different word.
  • Legends and figures have been mixed-up. Please check

Reviewer 2 Report

This manuscropt describes a study whose aim was to study the effect of microgravity on dermal fibroblasts, in particular with their ability to transdifferentiate towards the myofibroblastic phenotype, interact with collagen and keratinocytes when co-cultured in a 3D system. The study confirms the results of other authors on the effect of microgravity on fibroblast behavior, shows some interesting results on adhesion molecules and structures as well as on the cross-talk between fibroblasts and Keratinocytes, suggests hypotheses about the mechanisms underlying the observed effects.

Overall, the manuscript deals with an interesting topic but needs improvement in order to be published:

  • English needs revision
  • In materials and methods some procedures needs to be explained in more detail:

-explain why the angular velocity of rotation was set at 90/s as maximum and 30/s as minimum in a random mode and why you did not select a specific value of angular velocity (often 60/s is used)

-in the description reported in “materials and methods” it seems that only fibroblasts are exposed to microgravity, while in “results” and “discussion” it is reported that the co-culture with fibroblasts and keratinocytes is exposed to microgravity. The procedures for the exposure of the co-cultures to microgravity should be reported in “materials and methods”.

3)     In the copy I received, it seems that from Fig. 6 onwards the captions do not match, this makes it difficult to observe and understand the figures.

4)     More important: in my opinion there is an error in the design of the experiment.

In fact, the authors study the effect of microgravity on different functions of fibroblasts, then they expose co-cultures of fibroblasts and keratinocytes to microgravity and observe that the interaction between the two types of cells is altered in microgravity. Then, in the discussion they write "weightlessness- induced changes in fibroblasts involve concomitant modifications in

keratinocytes and in their mutual relationships when the two cell types are co-cultured in

a 3D-model " but they did not study the probable concomitant modifications induced by microgravity in keratinocytes.

Wanting to limit the study to the effects induced by microgravity in the behavior of fibroblasts only, it would have been more logical to observe the behavior of co-cultures held at 1G, but prepared starting from fibroblasts previously exposed to microgravity and keratinocytes always kept at 1G. In this way, the consequences of the microgravity-induced altered behavior of fibroblasts in the cross-talk with keratinocytes would have been clearer.

5)     In the discussion, the authors report that in co-cultures exposed to microgravity the distribution and localization of fibroblasts and keratinocytes differ from controls, but a clear hypothesis of the possible causes of the altered interaction between the two cell types is not advanced.

Moreover, it is well known that both fibroblasts and keratinocytes are mechanosensitive cells and their cross-talk is regulated by mechanical forces, interaction with ECM and production of soluble factors which can control the reciprocal behavior of the two cell types. Moreover, mechanical factors can affect the production of soluble factors by the cells themselves. These aspects should be discussed in the manuscript.

Round 2

Reviewer 1 Report

Author only partially answered to comments resulting from the evaluation of the first draft of the manuscript.

I can understand that Authors are not able to perform additional experiments, however, to make more consistent their data, they should select the data they want to present or, alternatively, they must clearly explain the significance of data obtained at time points which are not uniform along their experiments. The text has to be revised not only by adding few sentences in the discussion, but looking more carefully at the results and at their description.

Comments:

  • The major issue is that different experiments were performed at different time points, and not all data can be therefore compared or can consistently support Author’s conclusions. Observations done at 24 and 72 h or at 24 h+24 recovery are of interest, since they indicate that changes observed after 24h are transient. Changes in invasion/migration, alfa-SMA, cofilin and Fascin disappear in fact at 72h or after 24 h recovery. These changes are for instance correlated with a different number of cells? Changes in proliferation/apoptosis were not evaluated at 72h. Are all changes observed at 24h transient? If cells were investigated for some parameters after 72 hour, Authors should know at least the number of cells and at least these data can be added.
  • Authors simply changed the number of donors from 1 to 5. However, they did not explain, since they work with cells from different donors, if these cells were pooled or if they were kept separate. When authors state that experiment were done three times and in duplicate/triplicate, this means that experiments were three times and in duplicate/triplicate for each cell line? Or cells from different donors were used for different experiments. Authors should provide this information, which is important for a better and a more appropriate interpretation of results.
  • Figure 10: it not specified in the legend, neither in the text or in Materials and methods, for how many hours cells were treated. I suppose 24 h. Please specify
  • Collagen-gel contraction: Authors state that reduction of a-SMA and cofilin indicate “a de-differentiantion from myofibroblasts to quiescent fibroblasts”. Is this statement resulting from their data?. Since experiments were performed on 24h using fibroblasts, it is unlikely that cells in 24 h have the time to differentiate and to dedifferentiate.
  • Alpha-SMA expression. Authors show by WB a significant reduction of-SMA expression after 24 h, however, by confocal microscopy, it is clearly evident that differences are due to a different localization, i.e. translocation from cytoplasm to nucleus.  Changes in the cellular compartment do not mean that the global amount of protein (as measured by WB) was modified. Authors should comment these findings.
  • In the discussion, Authors, in the light of previous and present data, state that “new relationships between cells will drive the system to a different architecture”. It is hardly to say that the “architecture” is modified on the basis of changes observed only after 24 h. Since none of the parameters measured at 72 was modified, it can be hypothesized that even the interactions between keratinocytes and fibroblasts would be different and therefore Authors cannot speak of “architecture” but simply of cell-interactions.
  • Actin expression: no comments are provided for figure 8, whereas a reduction is observed in figure 9 (upper panel). Please check. Moreover, in figure 9 (lower panel) Authors state that there are no differences. Looking at the images and especially those in panels C and c they appear strikingly different. Authors must comment
  • Legend to figure 9: the measure unit of scale bar is missing for upper panels
  • Scale bars are not provided in the legend to figure 8.
  • In WB experiments and in MMP activity measurements samples were normalized for the same amount of protein/cells. Since the amount of cells was reduced, this aspect has to be specified in order to better evaluate the observed changes.
  • Authors should check for English errors.

Reviewer 2 Report

English was not improved

The response to Point 1 is not satisfactory: the authors write to the reviewer: "We apologize for the uncorrect report. Indeed, we established 60/s angular velocity on average. However, the system experienced, albeit unfrequently, some changes (i.e., from 30 to 90/s) when set in a random mode."

But they did not report this correction on the new version of the paper. Moreover, if really they establish 60/s angular velocity but sometimes the value changes from 30 to 90/s out of control, this could be a bias for the experiment and makes uncertain the collected data.

The authors answered to point 4 citing their previous paper on the behavior of keratinocytes exposed to microgravity. The paragraph they added in the paper only partially answered to my question, which was also methodological.

Round 3

Reviewer 1 Report

Authors revised the text according to most of reviewer's suggestions.

One point is still missing: concerning Point 2 Authors explained that cells from different donors were pooled together, but they did not add this explanation in the text.

Please add to the text the sentence you  have provided in  your reply "The cells from different donors were pooled all together. Experiments in duplicate/triplicate were performed using this same pool"

Author Response

Thank to the reviewer for the suggestion.

We added the following sentence "The cells from different donors were pooled all together. Experiments in duplicate/triplicate were performed using this same pool" in "Cell Culture" paragraph in Matherial and Methods section.